# A Reasonable Effectiveness of Features in Modeling Visual Perception of User Interfaces

**Maxim Bakaev** [1,*] **, Sebastian Heil** [2] **and Martin Gaedke** [2]

1   Independent UX Consultant, Novosibirsk 630132, Russia
2   Department of Computer Science, Technische Universitat Chemnitz, 09111 Chemnitz, Germany
*   Correspondence: maxis81@gmail.com

**Abstract:** Training data for user behavior models that predict subjective dimensions of visual perception are often too scarce for deep learning methods to be applicable. With the typical datasets in HCI limited to thousands or even hundreds of records, feature-based approaches are still widely used in visual analysis of graphical user interfaces (UIs). In our paper, we benchmarked the predictive accuracy of the two types of neural network (NN) models, and explored the effects of the number of features, and the dataset volume. To this end, we used two datasets that comprised over 4000 webpage screenshots, assessed by 233 subjects per the subjective dimensions of Complexity, Aesthetics and Orderliness. With the experimental data, we constructed and trained 1908 models. The feature-based NNs demonstrated 16.2%-better mean squared error (MSE) than the convolutional NNs (a modified GoogLeNet architecture); however, the CNNs' accuracy improved with the larger dataset volume, whereas the ANNs' did not: therefore, provided that the effect of more data on the models' error improvement is linear, the CNNs should become superior at dataset sizes over 3000 UIs. Unexpectedly, adding more features to the NN models caused the MSE to somehow increase by 1.23%: although the difference was not significant, this confirmed the importance of careful feature engineering.

**Keywords:** user interfaces; aesthetics; visual complexity; deep learning; neural networks

## 1. Introduction

We are now used to the abundance of data, of which the worldwide amount is already measured in zettabytes ($10^{21}$). Data is a primary resource for the current boom of AI/ML methods (the "AI summer"), but human-generated data can still be scarce.

In the field of human–computer interaction (HCI), the richest data deposit is interaction logs. In a broad sense, interaction logs may involve not just web pages visited or mouse clicks made [1], but also the user's eye movements or brain electrical activity [2]. However, we still struggle to reliably connect capturable manifestations in the human body to sophisticated processes in the human mind: for instance, facial expression recognition, even in combination with electroencephalography (EEG), achieves about 70% classification accuracy in emotion recognition [2]. Thus, HCI still heavily relies on collecting user-subjective impressions via surveys or dedicated experiments. These methods make the data somewhat costly, so the corresponding datasets in the field are generally limited to $10^3$ s of records.

One way in which HCI could benefit from rich data is by facilitating test automation, which has already demonstrated success in software engineering—in particular, by reducing costs for unit, and integration testing; however, GUI test automation has shown little progress, as some challenges have remained unresolved for almost two decades, as noted in the systematic literature review by Nass et al. (2021) [3]. Among the widely reported "accidental" challenges—being, as per F. Brooks' classification, the challenges that in principle can be resolved some time—are *C3: Robust identification of GUI widgets ("correct widget*

*is...that the human tester...would select in a given situation")* and *C19: Creating/maintaining model based tests*: these, at least—and the former is said to also affect another four major challenges—could evidently be helped by properly arranged user behavior data.

By "properly arranged", we mean user behavior models (UBMs) that can act as test oracles in dynamic GUI testing (static testing is already reasonably well-covered by code validators, snapshot testing, etc.). In the context of HCI, UBMs are formal constructs that conceptualize, explain or predict the behavior of human users with respect to particular user interface (UI) designs, without involving actual users: in the case of the latter, with no need to have a user, let alone make the user wear expensive equipment (cf. EEG-based emotion recognition), this can be a considerable advantage for rapid software development [4]. Process-wise, manual dynamic GUI testing is even more detrimental than general software testing, as external resources (target users) are needed to perform the task.

The considered UI is usually one of the UBM inputs—as code, screenshot, model or prototype, etc.—or a set of representative parameters describing it. Strictly speaking, UBMs do not have to be based on data, and such rule-based models are already used in the industry. A test case in Selenium (e.g., for a login web form) describes the behavior of an ideal user in ideal circumstances. Nevertheless, data are generally required to predict more user-specific or subjective aspects of interaction: these may involve dimensions of usability [1], aesthetics of webpage layouts [5], subjective impressions of visual complexity, aesthetics, and orderliness [6], etc. Some models can rely on log data, such as [1], which used about 23 GB of mouse and scrolling tracking data, but others have to deal with smaller datasets.

A notable example is Webthetics DNN [7], which used a deep learning (DL) neural network (NN) architecture for a dataset from 2013, containing approximately 400 webpage screenshots: it achieved a linear correlation corresponding to $R^2 = 72.3\%$, by relying on an ML model pre-trained on 80,000 general images. Indeed, for the aesthetic prediction of images, up-to-date DL models can boast an accuracy of over 80%, though with considerable variation, due to the NN architecture [8]. In the work done by Zhang et al. (2021), the accuracy also varied due to the dataset size:

- the *AVA dataset* had over 200,000 training and validation images, and the considered architectures demonstrated accuracy from 74.2% to 86.7%;
- the *Photo.net dataset* had about 12,500 training and validation images, and the considered architectures demonstrated accuracy from 59.9% to 81.0%.

Nevertheless, the mainstream approach to coping with limited dataset sizes (e.g., in predicting UI aesthetics)—indeed, the most popular dimension in "emotional user experience"—has been to rely on feature-based learning. Feature engineering in the field has been associated with the development of algorithms for extracting various quantitative parameters from UIs—the so-called UI metrics [9–12]. The resulting models explained about 65% of the variation in the surveyed visual complexity (cf. [9,12]), and about 50% of the variation in aesthetic perception (cf. [10] in [13], Table 15). The feature engineering approach has recently culminated in a model with as many as 384 features, used to predict webpage layout aesthetics [5]: the achieved accuracy was 87.9%, but for two levels only (classification), whereas, typically, UI aesthetics studies employ a 7-point Likert scale [13] (Table 8) (regression).

In summary, feature-based approaches still have their place in the visual analysis of "HCI vision" for GUI evaluation, even though the general computer vision moved to DL some time ago. In our work, we investigated which approaches yielded better results for user behavior modeling on a training dataset that was typically-sized for HCI. Our preliminary results were presented at the HCII 2022 conference, and were published as [14]. In the current paper, we additionally explored the effect of using more metrics as features in NN models. To that end, we extended the volume of the datasets that we use (from about 2500 to about 4000, with the corresponding increase in the number of human annotators), and the number of the models trained and analyzed (from 378 to 1380). We formulated and addressed the following research questions:

**RQ1:** Does feature engineering or automated feature extraction work better?
**RQ2:** What is the effect of having more data?
**RQ3:** What is the effect of using more features?
**RQ4:** Does it matter which subjective visual impression we are predicting?

The rest of our paper is structured as follows: in Section 2, we provide an overview of feature-based and DL approaches, with a focus on their application in HCI, as well as the tools for extraction of UI features (metrics); in Section 3, we provide the description of our experimental study, and the construction of the NN models, trained on two datasets of 4463 webpage screenshots, and assessed by 233 subjects; in Section 4, we use statistical analysis to compare the models' different architectures, and the number of features; we also consider the effect of the particular visual perception scales; in the final section, we summarize our contributions, discuss limitations and compare the obtained results to the ones presented in the existing literature.

## 2. Methods and Related Work

Neural Networks used to be based on features that required careful engineering, and thus relied on domain expertise; however, such "traditional" artificial neural networks (ANNs) came to be inferior at processing multidimensional data, such as videos, sound, etc., in their raw form. As the volumes of available and labeled data increased, DL models that could extract features automatically became the mainstream in many ML tasks, including image recognition. Graphical UI is a design artifact whose goal is to facilitate interaction; whether graphical UI could be considered an image, with respect to visual perception, remains an open question [15], which may well define the scope of effective methods. Thus, we overviewed both feature-based and DL methods, with a focus on their application to HCI.

### 2.1. Feature-Based UI Analysis

Naturally, UI analysis for static testing can be performed based on UI code. The effectiveness of code-based approaches has been demonstrated in detecting problems with HTML/CSS, URIs, accessibility, and even certain aspects of usability ("usability smells" [16]), etc. The main challenge is having a proper test oracle and comparator: thus, automatically checking content for broken URIs is straightforward (through HTTP status codes), while validating semantic correctness is not. In the thriving snapshot testing, a commonly employed test oracle is an image screenshot of the same UI, the correctness of which has been verified.

Fully dynamic GUI testing, however, implies interaction, so it is seldom, if ever, done with code. Visual (image-based) UI analysis offers the potential for extracting the features meaningful for user experience, such as perception of complexity, aesthetics, and regularity of layouts: for instance, "bad smells" can be violations of visual design guidelines, detected based on selected features extracted from UI images [17]. The quantitative characteristics of UIs are often called *metrics*, and a multitude of automated tools for their extraction has been developed over the past decade (including [11,18–21]). Research in the field has delivered quite a large number of metrics of varying relevance and predictive power, "fragmented across disciplines, UI types, data formats, and research groups" [11].

Aalto Interface Metrics (AIM), presented by Oulasvirta et al. (2018), was probably the first endeavor to gather metrics from different development teams in a single tool (https://interfacemetrics.aalto.fi/, accessed on 5 December 2022). The metrics were organized into several categories: color perception, perceptual fluency, visual guidance, and accessibility [11]. However, obtaining metrics for an individual UI screenshot has rather limited usefulness for research purposes. Another meta-tool, WUI Measurement Platform (http://va.wuikb.info/, accessed on 5 Deccember 2022) can collect the metrics provided by various external services, and save them in a database, for subsequent export in an ML-friendly data format [12]. The tool can use a list of many screenshots (filenames) as the input, thus implementing a "batch mode". The WUI Measurement Platform is described

in greater detail in a subsequent section of the current paper, as we used it for the metrics extraction in our experimental study.

Many algorithms for calculating metrics imply recognition of UI elements as an intermediate step: in this regard, a UI screenshot is recognized as a special item. The disadvantages of directly using some DL methods from computer vision for detecting objects in UIs were noted in [22], which also identified several specific characteristics of GUIs and GUI elements: high cross-class similarity; large in-class variance; packed or close-by elements and mix of heterogeneous objects. Nevertheless, recent DL methods have been successfully applied, particularly in regard to larger datasets that rely on interaction logs: for instance, it was demonstrated on the *Rico dataset*, which incorporated thousands of UI images, that a Deep AutoEncoder can be successful in the detection of elements in GUIs [23]; therefore, in the following subsection, we overview the application of DL approaches to visual analysis of UIs.

### 2.2. Deep Learning for UI Analysis

Mathematicians note that the architecture of deep NNs implies many hidden layers, each with a relatively small number of neurons, which makes them superior approximators for real tasks [24]. Moreover, some researchers say that the visual cortex in animals is organized similarly to this architecture [25], which explains its effectiveness in image recognition, classification, segmentation, etc.

The subjective opinions on aesthetic attractiveness, visual complexity and other visual perception dimensions are classically collected on a Likert scale [9,10]. As the scale is ordinal, and the ratings are often averaged between several assessors, the prediction is done as a regression rather than a classification. It has been also demonstrated that CNN models can work well in regression tasks, and predict human behavior and subjective judgments—for instance, image quality opinion scores [26]. In a recent study of aesthetic preferences for UI layouts, over 12,000 UI design images were collected from UI.cn and DOOOR.com, and were used in six deep CNNs (no $R^2$s were reported) [27]. The increased dataset size was facilitated by the use of the ratio between *likes* and *views*, provided by the design presentation platforms as the ground truth. Recognizably, this variable has a notable conceptual gap with individual aesthetic impressions, but the practical purpose of the latter is indeed to make a user "like" a UI visual appearance.

Our overview of the work in the field suggests that the following factors matter for the predictive accuracy of the models:

- **What is being predicted.** Even among the subjective dimensions of user experience, the achieved accuracies differ: e.g., over 60% for visual complexity, but only about 50% for aesthetic impressions. In [6], it was demonstrated that it also matters what domain a website relates to (culture, health, food, e-government, etc.)—especially for aesthetic assessment.

- **Available volumes of data.** Data have been credited with "unreasonable effectiveness" in ML; however, as explained above, HCI is not entirely satisfactory. In addition, some studies [28,29] suggest that the amount of data has a logarithmic relationship with the accuracy of DL models, which has seemingly been confirmed in practice: in [8], on a dataset about 16 times larger, the increase in the models' accuracy was about 1.11 times. Considering that the cost of obtaining additional data via surveys increases almost linearly, one might want to start looking for more reasonable effectiveness.

- **Architectures and features.** The current race to propose and polish NN architectures in ML suggests that this factor plays an important role. HCI is no exception: for instance, in [27] (Tables 4 and 5) one can see that the mean squared error (MSE) in UI layouts aesthetic preferences prediction varies considerably, due to NN architectures. Previously, we have highlighted that DL models need data—but the reverse is also true: to make use of bigger datasets, one generally needs to rely on more sophisticated models [29].

Correspondingly, the field faces a challenge in its development: *should it focus on selecting more meaningful metrics, and on polishing the respective algorithms, or on creating larger and higher quality datasets?* In our work, we benchmarked feature-based (ANN) and deep (CNN) models, in predicting several subjective dimensions of visual perception, based on a practically sized dataset. The methodology was reasonably straightforward, as inspired by [30]: "Classifiers are applied on raw and processed data separately and their accuracy is noted". In the following section, we describe the experimental study.

## 3. Experiment Description

### 3.1. Data, Design and Hypotheses

As user behavior models in HCI are domain-dependent [6], we sought diversity in our training data. Two datasets were employed in our study:

1.  *Dataset 1:* 2932 screenshots of website homepages. Each of the screenshots belonged to one of the six domains, as shown in Table 1. We collected the screenshots all at once, in 2018, first presented them in [6], and have now used them to study the effect of the NN architectures.
2.  *Dataset 2:* 1531 screenshots of website homepages, belonging to one of the eight domains. We merged this dataset from several very different sub-datasets collected by different researchers throughout the past decade (see Table 2). We used this more diverse dataset (smaller average number of websites per domain) to explore the effect of the number of features (UI metrics).

**Table 1.** Dataset 1: the screenshots collected from the six domains, and used in the models (with the drop-out due to the metrics calculation failure).

| Domain | Description | Number of the Screenshots | |
| --- | --- | --- | --- |
| | | Collected | Used |
| *Culture* | Websites of museums, libraries, exhibition centers and other cultural institutions. | 807 | 746 (92.4%) |
| *Food* | Websites dedicated to food, cooking, healthy eating, etc. | 388 | 369 (95.1%) |
| *Games* | Websites dedicated to computer games. | 455 | 362 (79.6%) |
| *Gov* | E-government, non-governmental organization and foundation websites. | 370 | 346 (93.5%) |
| *Health* | Websites dedicated to health, hospitals, pharmacies and medicaments. | 565 | 541 (95.8%) |
| *News* | Online and offline news editions' websites. | 347 | 328 (94.5%) |
| **Total:** | | **2932** | **2692 (91.8%)** |

**Table 2.** Dataset 2: the screenshots collected from eight sub-datasets, and used in the models (with the drop-out due to the metrics calculation failure).

| Domain/ Sub-Dataset | Description | Year | Resolution, px | Number of Screenshots | |
|---|---|---|---|---|---|
| | | | | **Nominal** | **Used** |
| *AVI_14* | From [10] | 2014 | W: 1278-1294 H: 799-800 | 140 | 124 (88.6%) |
| *Banks* | Screenshots of banks' websites * | 2022 | W: 1440 H: 960 | 304 | 287 (94.4%) |
| *CHI_15* | From [31] | 2015 | W: 1280 H: 800 | 75 | 68 (90.7%) |
| *CHI_20* | From [32] | 2020 | W: 720 H: 500-800 | 262 | 241 (92.0%) |
| *E-Commerce* | Screenshots of e-commerce websites * | 2022 | W: 1440 H: 960 | 156 | 148 (94.9%) |
| *English* | From [9] | 2013 | W: 1018-1024 H: 675-768 | 350 | 303 (86.6%) |
| *Foreign* | From [9] | 2013 | W: 1024 H: 768 | 60 | 51 (85.0%) |
| *IJHCS* | Part of the dataset from [33] via [32] | 2012 | W: 1000 H: 798-819 | 184 | 149 (81.0%) |
| **Total** | | | | **1531** | **1371 (89.5%)** |

* Provided by A. Miniukovich within the framework of the project FWF M2827-N.

To reflect the distinct dimensions of users' visual perception, we used three subjective scales: *Complexity*, *Aesthetics* and *Orderliness*. Each of the scales was represented as a Likert scale, ranging from 1 (lowest) to 7 (degree). The corresponding ratings, averaged per the assessments, were the output in the models of both architectures. The inputs were different, being the normalized metrics obtained for the screenshots for the ANN models, and the resized screenshot images for the CNN models.

We used a partial within-subject design for our experiment, relying on the same training datasets for the different architectures and the different numbers of features. To increase the statistical power of the comparisons, and to explore the effect of the dataset sizes, we trained the models, using all possible combinations of the sub-datasets in each dataset (e.g., in Dataset 1: News, News + Food, Culture + Gov + Health, etc.). With Dataset 1, we trained $2^6 - 1 = 63$ models for each of the three subjective impression scales, and for each architecture. With Dataset 2, we trained $2^8 - 1 = 255$ models for each scale, and for each considered number of features.

In line with the usual practice for NN models that perform regression tasks, we operationalized the quality of the models as the mean squared error (MSE) on the testing set (20% of the samples):

$$MSE = \frac{1}{n} \sum_{i=1}^{n} (y_i - \hat{y}_i)^2 \tag{1}$$

where $\hat{y}_i$ was the predicted value and $y_i$ the true value.

Our main goal was to compare the predictive ability of the feature-based ANNs (with additional exploration of the features) and the deep CNNs (with additional exploration of the dataset size factor); thus, we employed the following independent variables:

- The NN model's **architecture**: *Architecture* ∈ {ANN/CNN};
- The subjective visual impression **scale**: *Scale* ∈ {Complexity/Aesthetics/Orderliness};
- The **number of features** in the ANN models (detailed in the next sub-section), *Features* = 32 (resulting in *ANN32* models) or *Features* = 42 (resulting in *ANN42* models).

The size of the training dataset for the ANN and the CNN (Dataset 1) became the additional derived independent variable, *N*, varying in the range of $263 \leq N \leq 2154$.

The models were the dependent variables, while the derived dependent variables actually used in the study were the MSEs and the training times for Dataset 1 (longer training times for *ANN42* compared to *ANN32* were evident):

- *MSE-ANN* and *MSE-CNN*;
- *Time-ANN* and *Time-CNN*;
- *MSE-ANN32* and *MSE-ANN42*.

The null hypotheses were as follows:

**H$_0$1:** *There is no difference in MSE due to Architecture.*

**H$_0$2:** *The outcome of H$_0$1 is not affected by N.*

**H$_0$3:** *There is no difference in MSE per Features.*

**H$_0$4:** *The outcomes of H$_0$1 and H$_0$3 are the same for any Scale.*

### 3.2. The Material and the Models' Input Data

To collect the material for the Dataset 1, we asked student volunteers to supply us with URIs for homepages of websites belonging to the specified different domains. Then, our dedicated script would render the webpages, and capture them as $1280 \times 960$ or $1280 \times 900$ images, which would further undergo manual inspection. The selection of the websites, and the screenshots collection process, are described in more detail in one of our previous studies [6].

To assemble a more diverse Dataset 2, we now turned to several historical HCI datasets. The included websites were not just from different domains, but also from different years, starting a decade ago. The screenshots also had different resolutions; however, unlike the CNN models, the metrics extraction did not generally require uniform resolutions or aspect ratios (although some services may have cut or resized the input data).

The screenshots from both Dataset 1 and Dataset 2 were submitted to the WUI Measurement Integration Platform in "batch" mode. For each screenshot from Dataset 1, we extracted 32 metrics, whereas for the ones from Dataset 2, we extracted the same 32 metrics plus 10 additional ones (Table 3). The metrics were provided by two services: VA and AIM (cf. [12]).

For 240 (8.19%) screenshots in Dataset 1, and for 160 (10.45%) screenshots in Dataset 2, the services did not produce the full set of the metrics: these 400 screenshots were excluded from the datasets—even for the CNN, for the sake of comparison with the ANN. Although this dropout was not random, the domains were affected nearly equally (see the "Used" columns in Tables 1 and 2). The metrics' values were subsequently used as the input data in the feature-based models' training and testing for:

- *ANN models*, with reduced Dataset 1 (2692 screenshots), and the basic 32 features;
- *ANN32 models*, with reduced Dataset 2 (1371 screenshots), and the basic 32 features;
- *ANN42 models* with reduced Dataset 2, and the 32 basic plus 10 additional features.

**Table 3.** The 32 basic and the 10 additional metrics used for the ANN models.

| Service–Category | Metric |
|---|---|
| **The basic 32 metrics** | |
| Visual Analyzer (VA) quantitative | PNG filesize (in MB) <br> JPEG 100 filesize (in MB) <br> No. of UI elements <br> No. of UI elements' types <br> Visual complexity index |
| AIM Color Perception | Unique RGB colors <br> HSV colors avg Hue <br> HSV colors avg Saturation <br> HSV colors std Saturation <br> HSV colors avg Value <br> HSV colors std Value <br> HSV spectrum HSV <br> HSV spectrum Hue <br> HSV spectrum Saturation <br> HSV spectrum Value <br> Hassler Susstrunk dist A <br> Hassler Susstrunk std A <br> Hassler Susstrunk dist B <br> Hassler Susstrunk std B <br> Hassler Susstrunk dist RGYB <br> Hassler Susstrunk std RGYB <br> Hassler Susstrunk colorfulness <br> Static clusters <br> Dynamic CC clusters <br> Dynamic CC avg cluster colors |
| AIM Perceptual Fluency | Edge congestion <br> Quadtree Dec balance <br> Quadtree Dec symmetry <br> Quadtree Dec equilibrium <br> Quadtree Dec leaves <br> Whitespace <br> Grid quality (no. of alignment lines) |
| **The additional 10 metrics** | |
| AIM Color Perception | LABcolors (mean Lightness) <br> LABcolors (std Lightness) <br> LABcolors (mean A) <br> LABcolors (std A) <br> LABcolors (mean B) <br> LABcolors (std B) <br> Luminance std |
| AIM Perceptual Fluency | edge density <br> FG contrast <br> pixel symmetry |

*3.3. The Subjects and the Models' Output Data*

The assessments of the websites per the three subjective scales were provided in two dedicated surveys (with about 2.5 years between them) of two groups of subjects:

1.  For Dataset 1: 137 participants (67 female, 70 male), of ages ranging from 17 to 46 (mean 21.18, SD = 2.68). The majority of the participants were Russians (89.1%),

2.     the rest being from Bulgaria, Germany and South Africa. In total, the participants provided 35,265 assessments for the 2692 screenshots from the six domains.

2.     For Dataset 2: 96 participants (27 female, 69 male), of ages ranging from 19 to 25 (mean 21.02, SD = 1.30). The majority of the participants (93.8%) were Russian, with the others representing Uzbekistan. In total, the participants provided 24,114 assessments for the 1371 screenshots from the eight domains.

Most of the participants were Bachelor's and Master's students, but some were office workers employed in IT. All of the subjects took part in the experiment voluntarily, and had normal or corrected-to-normal vision.

The assessment of the screenshots from both datasets was performed by our dedicated online survey tool. More details on the assessment procedure can be found in [6]. The outcome of the assessments per the two datasets is presented in Table 4.

**Table 4.** The subjective visual perception scales for the two employed datasets.

| Domain/ Sub-Datasets | Complexity | | Aesthetics | | Orderliness | |
|---|---|---|---|---|---|---|
| | **Mean** | **SD** | **Mean** | **SD** | **Mean** | **SD** |
| **Dataset 1** | | | | | | |
| *Culture* | 3.63 | 0.81 | 4.24 | 0.99 | 4.29 | 0.90 |
| *Food* | 3.66 | 0.81 | 4.70 | 0.94 | 4.66 | 0.87 |
| *Games* | 3.57 | 0.93 | 4.24 | 1.14 | 4.33 | 1.00 |
| *Gov* | 3.81 | 0.82 | 3.86 | 0.92 | 4.14 | 0.86 |
| *Health* | 3.73 | 0.79 | 4.15 | 0.90 | 4.40 | 0.82 |
| *News* | 4.16 | 0.86 | 3.80 | 0.83 | 4.16 | 0.82 |
| **Total** | **3.71** | **0.83** | **4.17** | **0.98** | **4.34** | **0.86** |
| **Dataset 2** | | | | | | |
| *AVI_14* | 3.15 | 0.85 | 4.15 | 1.15 | 4.32 | 0.97 |
| *Banks* | 3.20 | 0.78 | 4.33 | 0.95 | 4.81 | 0.81 |
| *CHI_15* | 3.70 | 0.84 | 3.44 | 1.06 | 3.91 | 0.93 |
| *CHI_20* | 3.52 | 0.77 | 3.62 | 0.96 | 4.14 | 0.73 |
| *E-Comm* | 2.93 | 0.62 | 4.38 | 0.93 | 4.63 | 0.75 |
| *English* | 3.68 | 0.93 | 3.05 | 0.94 | 3.78 | 0.83 |
| *Foreign* | 4.29 | 1.01 | 2.75 | 0.75 | 3.37 | 0.86 |
| *IJHCS* | 3.88 | 1.00 | 2.83 | 0.91 | 3.68 | 0.82 |
| **Total** | **3.47** | **0.91** | **3.65** | **1.13** | **4.18** | **0.93** |

### 3.4. The ANN and CNN Models

As explained previously, we trained the following numbers of models:

- *ANN*: $3 \times (2^6 - 1) = 189$ models;
- *CNN*: $3 \times (2^6 - 1) = 189$ models;
- *ANN32*: $3 \times (2^8 - 1) = 765$ models;
- *ANN42*: $3 \times (2^8 - 1) = 765$ models.

To construct and train the ANN and CNN models, we used the `Colab` service freely offered by Google (TensorFlow 2.5 environment with Keras 2.4). The feedforward ANN models were built with the `Keras Tuner` library [34], which helped select the best models, with different combinations of various layers and hyperparameters. The models were trained until the verification accuracy began to decrease for several epochs in a row, i.e., a stopping mechanism was employed. The ANN models' code (for Dataset 1) was as follows:

```
def build_model(hp):
    model~=~keras.Sequential()
    activation_choice~=~hp.Choice('activation', values= ['relu',
        ↪ 'sigmoid', 'tanh', 'elu', 'selu'])
    model.add(Dense(units=hp.Int('units_input', min_value=512,
        ↪ max_value=1024, step=32), activation=activation_choice)
        ↪ )
    for i in range(2, 6):
        model.add(Dense(units=hp.Int('units_' + str(i), min_value
            ↪ =128, max_value=1024, step=32), activation=
            ↪ activation_choice))
    model.add(Dense(1))
    model.compile(optimizer="adam", loss='mse', metrics=['mse',
        ↪ coeff_determination])
    return model
```

The CNN models were built using GoogLeNet architecture, with the output layer being replaced with a single neuron layer to accommodate the regression task. We relied on the Adam optimization algorithm, and used a computer that had four i7-3930K CPUs @ 3.20 GHz, 16 GB of memory and NVIDIA Quadro RTX 5000.

## 4. Results

### 4.1. Descriptive Statistics

In total, we built 1908 models and registered *MSE* and *Time* for each of them; however, for Dataset 2 we discarded the models for which the training dataset size was less than 263, as non-valid: this was done for the sake of better comparison with the models trained on Dataset 1, and in line with the popular "rule of thumb" recommending no less than 10 data records per feature. In the further analysis, we used 690 models in ANN32 (90.2% valid) and 690 models in ANN42 (90.2% valid).

The Pearson correlation between the dependent variables' values obtained for the same combinations of domains turned out to be highly significant:

- between *MSE-ANN* and *MSE-CNN*: $r_{189} = 0.341$, $p < 0.001$;
- between *Time-ANN* and *Time-CNN*: $r_{189} = 0.623$, $p < 0.001$;
- between *MSE-ANN32* and *MSE-ANN42*: $r_{690} = 0.671$, $p < 0.001$.

According to the Shapiro–Wilk's tests, the normality hypotheses had to be rejected for *MSE-ANN* ($W_{189} = 0.901$, $p < 0.001$), *MSE-ANN32* ($W_{690} = 0.946$, $p < 0.001$) and *MSE-ANN42* ($W_{690} = 0.927$, $p < 0.001$), but not for *MSE-CNN* ($W_{189} = 0.989$, $p = 0.171$).

For Dataset 2, as expected, *Time-ANN42* (mean = 73.0, SD = 11.4) was slightly greater than *Time-ANN32* (mean = 71.4, SD = 10.1). The detailed descriptive statistics (means and standard deviations) for the main dependent variables are presented in Tables 5–7.

**Table 5.** Dataset 1: descriptive statistics for MSE, as per the architectures and the scales.

| Scale | MSE-ANN | MSE-CNN |
|---|---|---|
| *Complexity* | 0.644 (0.081) | 0.750 (0.127) |
| *Aesthetics* | 0.772 (0.104) | 0.968 (0.182) |
| *Orderliness* | 0.769 (0.102) | 0.859 (0.122) |
| **All** | **0.739 (0.106)** | **0.859 (0.170)** |

**Table 6.** Dataset 1: Descriptive statistics for Time, as per the architectures and the scales.

| Scale | MSE-ANN | MSE-CNN |
|---|---|---|
| *Complexity* | 93.7 (17.4) | 3265.4 (1440.7) |
| *Aesthetics* | 107.2 (20.1) | 2704.4 (1560.0) |
| *Orderliness* | 101.3 (18.6) | 2650.6 (1495.8) |
| **All** | **100.7 (19.4)** | **2506.8 (1511.3)** |

**Table 7.** Dataset 2: Descriptive statistics for MSE, as per the number of features and the scales.

| Scale | MSE-ANN32 | MSE-ANN42 |
|---|---|---|
| *Complexity* | 0.648 (0.122) | 0.654 (0.128) |
| *Aesthetics* | 0.864 (0.166) | 0.858 (0.168) |
| *Orderliness* | 0.654 (0.101) | 0.680 (0.141) |
| **All** | **0.722 (0.166)** | **0.731 (0.172)** |

*4.2. Dataset 1: The Models' Training Time*

We performed correlation analysis to check if the training yielded the models with the best possible quality (i.e., if the stopping mechanism acted correctly). The correlations between *Time-ANN* and *MSE-ANN* ($p = 0.140$), as well as between *Time-CNN* and *MSE-CNN* ($p = 0.104$), were not significant. As expected, the Pearson correlations between the training datasets' sizes and the training times turned out to be positive and highly significant for both the ANN ($r_{189} = 0.812, p < 0.001$) and the CNN ($r_{189} = 0.765, p < 0.001$).

In our experiment, an ANN model, on average, trained 25 times faster than a CNN one; however, the former required an initial "investment", in the form of the feature extraction process. With our WUI Measurement Platform, it took about 43 h for Dataset 1 (i.e., 57 s per one valid screenshot): this implies that, given the same number of screenshots and the training machine configuration, the CNN approach would need less overall time than the ANN, unless **64 or more models were constructed**.

For the models trained on Dataset 1, ANOVA suggests the significant effects of *Architecture* ($F_{1,371} = 701.1, p < 0.001$) and *Scale* ($F_{2,371} = 3.71, p = 0.025$) on *Time*. The interaction of the two factors was also significant ($F_{2,371} = 3.41, p = 0.034$). As the data presented in Table 6 also suggest, training a model for the *Complexity* scale was faster with the ANN, but slower with the CNN.

*4.3. Comparison of the Models' MSEs*

4.3.1. Dataset 1

Despite *MSE-CNN* not being normally distributed, we used a t-test for paired samples (equal variances not assumed), to check the difference in *MSE-ANN* and *MSE-CNN*. The difference turned out to be highly significant: $t_{188} = -9.904, p < 0.001$. Figure 1 demonstrates the differences in the distributions of the MSEs for the two architectures.

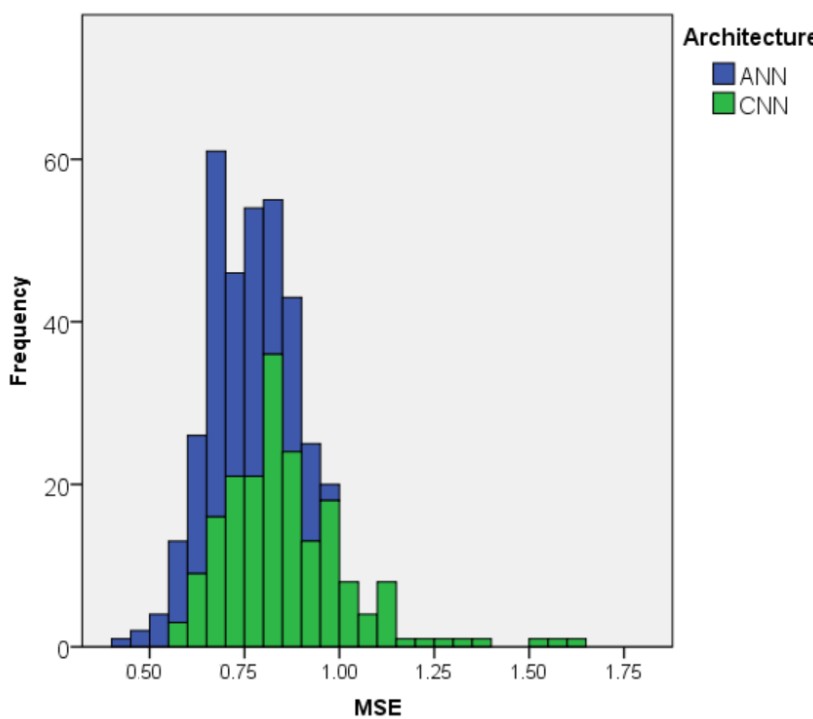

**Figure 1.** The distributions of the models' MSEs for the ANN and the CNN architectures.

Furthermore, we performed the analysis of the difference with all the factors, to adhere to the effect of scales. For Dataset 1, ANOVA found significant effects of *Architecture* ($F_{1,371} = 90.5, p < 0.001$) and *Scale* ($F_{2,371} = 53.1, p < 0.001$) on MSE, and the interaction between them was also significant ($F_{2,371} = 9.0, p < 0.001$). Post-hoc tests found significant differences (at $\alpha = 0.001$) between all three scales.

Notably, *MSE-CNN* had a significant negative Pearson correlation with N ($r_{189} = -0.174$, $p = 0.017$), which implies that the CNN models could benefit from the larger size of Dataset 1. Meanwhile, the correlation between *MSE-ANN* and N was not significant ($r_{189} = 0.033$, $p = 0.655$).

### 4.3.2. Dataset 2

Unexpectedly, the mean *MSE-ANN32* (0.722) was 1.23% **lower** than the corresponding error value for the models with more features, *MSE-ANN42* (0.731); however, the t-test for paired samples suggested that the difference was not significant ($t_{689} = -1.61, p = 0.107$). ANOVA testing with all the factors found a significant effect of *Scale* ($F_{2,1374} = 322.3$, $p < 0.001$), but not of *Features* ($F_{1,1374} = 1.26, p = 0.262$), nor interaction ($F_{2,1374} = 1.49$, $p = 0.227$).

### 4.4. Regression Analysis for MSE-CNN

To formalize the effect of N on *MSE-CNN*, we used regression analysis. The model's $R^2 = 0.03$ was low, but the regression turned out to be significant ($F_{1,187} = 5.85, p = 0.017$):

$$MSE\text{-}CNN \ = \ 0.932 - 0.633 \times 10^{-4} N \tag{2}$$

Stretching the model (2) beyond the interval used in our study ($N \leq 2153$), we can speculate that at $N = 2912$, *MSE-CNN* would catch up with the average *MSE-ANN* = 0.739 (the latter was not significantly affected by N, as we noted before).

As we previously found a significant effect of *Scale*, we built regression models with N for the difference between the MSEs. The models were significant for Aesthetics ($F_{1,61} = 3.99, p = 0.050, R^2 = 0.06$) and Orderliness ($F_{1,61} = 5.34, p = 0.024, R^2 = 0.08$), but not for Complexity ($F_{1,61} = 0.42, p = 0.520$):

$$(MSE\text{-}CNN - MSE\text{-}ANN)_{Aesthetics} = 0.315 - 1.082 \times 10^{-4}N \tag{3}$$

$$(MSE\text{-}CNN - MSE\text{-}ANN)_{Orderliness} = 0.188 - 0.900 \times 10^{-4}N \tag{4}$$

From (3), it follows that for the Aesthetics models, *MSE-CNN* would become smaller than *MSE-ANN* when $N > 2908$. For the Orderliness models (4), the same would be achieved when $N > 2090$.

Finally, we constructed linear regression models for MSE in Dataset 1, with dummy variables (i.e., having the values 0/1): $Scale_A$ had the value of 1 if the current model predicted Aesthetics, and $Scale_O$ had the value of 1 if the current model predicted Orderliness. Another dummy variable was $A_{CNN}$, which had the value of 1 if the current model was built using the CNN architecture. The rational scale factor in the regression was $N$, the training dataset size for the current model. All the variables turned out to be significant (at $\alpha = 0.05$) in the resulting model, which had $R^2 = 0.341$ ($F_{4,373} = 48.2$, $p < 0.001$):

$$MSE = 0.684 + 0.121A_{CNN} + 0.158Scale_A + 0.102Scale_O - 0.293 \cdot 10^{-4}N \tag{5}$$

## 5. Discussion and Conclusions

Deep learning models currently dominate most AI/ML sub-fields. For instance, deep convolutional neural networks (CNNs) are the de facto standard in computer vision, for performing image recognition, classification, etc. In some contexts, though, feature-based approaches still have their place in visual analysis, particularly as they are less "hungry" for data, e.g., in [35] they relied on a dataset of just over 5000 images, which is comparable to our study. In a classification task, they managed to achieve a success rate of about 95%, which is comparable with the ones reported for CNN-based approaches; however, the latter generally require more training data, e.g., 16,000 images in [36]. In our work, we focused on the ML-based user behavior models that are increasingly used in HCI—one of the fields that rely on expensive, human-generated data.

In particular, we compared feature-based ANNs operating with a varying number of UI features to CNNs that can learn from raw UI images. In total, we constructed 1908 models, and trained them on two datasets, which together incorporated over 4000 web UI screenshots. The output of the models were predictions of subjective user impressions, as per the Complexity, Aesthetics and Orderliness scales. In total, we employed 233 participants in our experimental study. The participants provided about 60,000 assessments for the screenshots per the three scales. Our analysis results suggest the following outcome for the formulated research questions:

**RQ1:** **Feature-based models worked better for our datasets.** There was significant difference ($p < 0.001$) in the models' MSEs per the architectures. The average MSE for the ANN architecture was 16.2% smaller than for the CNN architecture. There was also a significant difference in the models' training time per the architectures, assuming that the features for the ANNs were pre-extracted: training a single CNN model took 0.696 h, which was 25 times longer than for an ANN model; however, extracting the features for the 2692 screenshots in Dataset 1 took about 43 h.

**RQ2:** **The size of the training dataset had a positive effect on the CNN models' MSE.** $N$ significantly affected *MSE-CNN* ($p = 0.017$), but not *MSE-ANN* ($p = 0.655$): thus, at $N > 2912$, the former should have become smaller than the latter, although we did not observe it in our study, due to the limited size of Dataset 1. Naturally, $N$ also had a significant effect on the training time for both architectures.

**RQ3:** **The superficial increase in the number of features did not have a positive effect.** There was no significant difference in *MSE-ANN32* and *MSE-ANN42*, and even the average error was 1.23% better in the models with **fewer** features. Training the models with more features also took, on average 2.19%, more time. In accordance with the popular principle, features must not be multiplied beyond necessity: with respect to ML, the violation of this principle can lead to overfit and poor MSE on test data. On the other hand, if the increase in the number of features had been

accompanied by a corresponding boost in the training data volume, the result might have been different. Another consideration is that ML models and algorithms have a kind of "saturation point", after which they are not able to make good use of extra training data [29].

**RQ4:** **The particular dimension of the visual perception significantly affected the quality of the models; however, the ANN models were always superior to the CNN models.** For the different values of *Scale*, the outcome of the models' comparison varied: for Dataset 1, the Complexity models, on average, had 19.6% smaller *MSE-ANN* and 21.8% smaller *MSE-CNN*; moreover, *N* did not have a significant effect on the errors for Complexity, unlike for the other two scales; for Dataset 2, the average MSE for Aesthetics was even higher (32.2% worse than for Complexity), but the MSE for Orderliness had improved considerably. All in all, visual complexity appeared to be the most advantageous of the dimensions: thus, the widespread notion of using it as a mediator for aesthetic impression [32] appeared well-justified.

Unfortunately, no research publications came to our attention that explicitly compared different NN architectures and ML approaches in image-based UI analysis tasks; however, some existing work in text analysis [30] and document classification [37] appear relevant in validating our findings, as they operated with comparable dataset sizes. For instance, in [37], the authors similarly discovered that a feature-based approach with normalized difference measure had an advantage over DL for text documents classification. More generally, it has been demonstrated that feature-based tweaking can improve object detection in GUI [22]. Previously, we obtained comparable MSEs for ANN models trained on data that largely intersected with Dataset 1: on average, 0.928 for Complexity, 1.09 for Aesthetics and 1.119 for Orderliness [6]; however, we used a simpler NN architecture, and did not perform the hyperparameters optimization, which would explain the somehow inferior MSEs. Notably, in that experiment, we also could not find a significant effect of the training dataset size on the ANN models' MSE.

The MSE values that we obtained in the current study were rather high, so we discourage anyone from production use of the models. As our goal was to compare the architectures, we did not use the techniques that can considerably improve CNN models (cf. [7]): transfer learning; data augmentation, etc. Another reason behind the relatively high MSE values was the limited size of the datasets, which were, however, realistic for HCI. General DL computer vision models can usually benefit from datasets that are several orders of magnitude larger: for instance, the aesthetics of photographs were predicted with a set of 255,000 images in [38], and with a comparable number in [8].

It is probably that the ANN models likewise did not have the best obtainable MSE, primarily because we did not actually engineer the features, but relied on the metrics that the available services (VA and AIM) could produce. The services introduced an additional drop-out as, for about 10% of the website, screenshots VA and/or AIM would not output the full set of feature values: this could be considered an additional disadvantage of the feature-based approach, although in practice it could possibly be mitigated by debugging the metric-calculating algorithms.

We believe, nevertheless, that our study's validity was reinforced by the high number of models that we created, and the high statistical power of the results. We consider our study to have made the following main contributions, which may be useful for researchers and practitioners who apply AI methods in HCI:

1. We found that on HCI-realistic datasets that included several hundred websites, feature-based NN models were reasonably (16.2%) more accurate at predicting users' visual perception dimensions.
2. We provided an estimate of the training dataset size at which deep learning should start being advantageous: this estimation—about 3000 websites—assumed that the effect of *N* on *MSE-CNN* was linear.
3. We demonstrated that one should be reasonable with the number of features, as a superficial increase might actually be damaging for the ANN models.

4.  We found that the results could vary, depending on the concrete subjective dimension being predicted: Complexity was confirmed as the most opportune one, while Aesthetics was predictably the most evasive.
5.  We explored the time required for extracting the features and training the models. A deep learning approach that works with raw input data can save on the total time, only if several dozen models are built. Feature extraction means a considerable fixed time cost, but it enables better subsequent flexibility in exploring and tweaking the models.
6.  We formalized a regression model (5) for estimating MSEs, based on the available training dataset size, the chosen architecture and the subjective user visual perception dimension being predicted.

**Author Contributions:** Conceptualization, M.B. and S.H.; methodology, M.B.; software, S.H.; validation, M.B. and S.H.; formal analysis, M.B.; investigation, M.B. and S.H.; resources, M.G.; data curation, M.B. and S.H.; writing—original draft preparation, M.B.; writing—review and editing, M.B. and S.H.; visualization, M.B. and S.H.; supervision, M.G.; project administration, M.G.; funding acquisition, M.G. All authors have read and agreed to the published version of the manuscript.

**Funding:** This research was funded by the Deutsche Forschungsgemeinschaft (DFG, German Research Foundation)—Project-ID 416228727—SFB 1410.

**Institutional Review Board Statement:** The study was conducted according to the guidelines of the Declaration of Helsinki, and was approved by the Ethics Committee of the Faculty of Humanities of Novosibirsk State Technical University (protocol code 7_02_2019).

**Informed Consent Statement:** Informed consent was obtained from all subjects involved in the study.

**Data Availability Statement:** Software code involved in this study will be available in a publicly accessible repository https://github.com/heseba/leonid, accessed on 5 December 2022. The data presented in this study are available on request from the corresponding author. The data are not publicly available, due to possible restrictions of third parties, particularly for the website screenshots.

**Acknowledgments:** Special thanks to Vladimir Khvorostov, who provided programming support for the research project. We would also like to thank all the participants in our experimental studies, including our Bachelor and Master students.

**Conflicts of Interest:** The authors declare no conflict of interest.

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
