# Peer review of "A Reasonable Effectiveness of Features in Modeling Visual Perception of User Interfaces"

_2504-2289, doi:10.3390/bdcc7010030_

Round 1

Reviewer 1 Report

The goal of this paper is to compare the user behavior predictive ability of neural 216 network models with two different architectures: “classical” feature-based ANNs (with 217 additional exploration of the features) and “deep” CNNs.

Please explain why you did not consider classification algorithms in your study.

The proposed method and the obtained results should be compared with other similar studies in the literature in order to be validated.

Please explain how much the obtained results are affected by the size of the considered databases.

Please specify the degree of confidence of the results obtained and presented in the conclusion chapter and how the obtained results could be validated.

Author Response

We would like to thank the esteemed reviewers for the time and effort spent on considering our manuscript and for the provided recommendations that help improving our paper.

We have revised the manuscript, adding and modifying it according to the reviewers’ comments, and also incorporating the editor’s request to make it more distinct with our previous conference paper. The list of references has been modified accordingly.

Our responses to the reviewers’ comments follow.

Please explain why you did not consider classification algorithms in your study.

The subjective opinions on aesthetic attractiveness, visual complexity and other visual perception dimensions are classically collected on a Likert scale (Reinecke, 2013; Miniukovich, 2014). Since the scale is ordinal and the ratings are often averaged between several assessors, the prediction is done as a regression rather than classification. We have added this explanation to the revised version of the manuscript.

The proposed method and the obtained results should be compared with other similar studies in the literature in order to be validated.

Unfortunately, no research publications that explicitly compare different NN architectures and ML approaches in image-based UI analysis task came to our attention. However, some other fields similarly have to deal with small dataset sizes when building ML models, so we have revised and extended the Discussion and Conclusion section to include the comparison to these results.

Please explain how much the obtained results are affected by the size of the considered databases.

We are not sure what “databases” the esteemed reviewer refers to. Possibly, these should be “datasets”? Then this comment corresponds to our RQ2. Our results suggest that the dataset size (N) has negative correlation with MSE in CNN models, but not in ANN models. We use this to make a very preliminary prediction (assuming linearity of the effect) about the threshold where CNN becomes superior over ANN. Some studies suggest that the effect is logarithmic though, so we mention this in the Discussion.

Please specify the degree of confidence of the results obtained and presented in the conclusion chapter and how the obtained results could be validated.

The revised version of the manuscript includes the discussion of the obtained results, particularly with regard to MSE values. We explain some limitations of our study and some research question-specific assumptions done for both types of models, ANN and CNN.

Reviewer 2 Report

Overall, this paper is well written.  The language needs to be tightened up:

In the abstract:  visual perception is often scarce short of deep learning (unclear)

The second and third sentences in the first paragraph of the introduction are  unclear and refer to an "AI summer" that is neither common knowledge nor is it cited.

In many instances, sentences are started with colloquial language and need to be tightened up.  For example, the second paragraph of the introduction, begins "So, ..."  - why include the "So"?  Just remove it.  There are many other instances where phrases are included that add nothing to the explanation of the author's narrative.

In 2.1, "As we mentioned before, " - no need to include this phrase.  Just remove it.

The study is well-motivated and well-explained.  The methodology is covered in sufficient detail.

This paper could be improved by covering more explanation on why they got the results to their four research questions in section 5, particularly RQ3 since this finding was surprising (Hint:  Occam's razor)

This paper concludes nicely, and the paper's contribution is sufficient for publication.  However, some enhancement to the writing to tighten up the language and explain some items in more detail will make this a much better paper.

Author Response

We would like to thank the esteemed reviewers for the time and effort spent on considering our manuscript and for the provided recommendations that help improving our paper.

We have revised the manuscript, adding and modifying it according to the reviewers’ comments, and also incorporating the editor’s request to make it more distinct with our previous conference paper. The list of references has been modified accordingly.

Our responses to the reviewers’ comments follow.

In the abstract:  visual perception is often scarce short of deep learning (unclear)

The second and third sentences in the first paragraph of the introduction are  unclear and refer to an "AI summer" that is neither common knowledge nor is it cited.

In many instances, sentences are started with colloquial language and need to be tightened up.  For example, the second paragraph of the introduction, begins "So, ..."  - why include the "So"?  Just remove it.  There are many other instances where phrases are included that add nothing to the explanation of the author's narrative.

In 2.1, "As we mentioned before, " - no need to include this phrase.  Just remove it.

We have implemented the suggestions to improve readability in the revised version of the manuscript.

This paper could be improved by covering more explanation on why they got the results to their four research questions in section 5, particularly RQ3 since this finding was surprising (Hint:  Occam's razor)

We thank the esteemed reviewer for the comment and the hint. We have extended the discussion of RQ3 in the final section of the revised manuscript.